# Peer review of "Reorganization Tips from a Sarcoma Unit at Time of the COVID-19 Pandemic in Italy: Early Experience from a Regional Referral Oncologic Center"

_jcm, 2020, doi:10.3390/jcm9061868_

Round 1

Reviewer 1 Report

The authors present a well organized structure for their sarcoma unit, although there is no clear data on how those patients were affected and how their management has been amended and improved. yet, this paper could work and replicated for other cancer systems. I also recommend the authors to cite below paper as it discussed nicely the multi-organ system response to COVID, including immunosuppressed and cancer cohort. 

Zaim S, Chong JH, Sankaranarayanan V, Harky A. COVID-19 and Multi-Organ Response [published online ahead of print, 2020 Apr 28]. Curr Probl Cardiol. 2020;100618. doi:10.1016/j.cpcardiol.2020.100618

Author Response

The paper of Zaim S. et al has been cited in discussion (lines 296-301). The reference list has been changed as well as through the manuscript. Indications and concepts about new prioritizations of sarcoma patients has been stressed and compared with ESMO/EURACAN recommendations and guidelines in the paragraph “Sarcoma unit approach”. The concept of amelioration of our management during COVID era has been highlighted in the new version of the manuscript through the integration with adjuvant treatment and by having guaranteed treatment for urgent, high risk cases of musculoskeletal malignancy.

Reviewer 2 Report

I enjoyed reading this and found it worthwhile seeing how you managed this pandemic. I think this should be published as it is likely this will not be our last pandemic.

It would be helpful for non-Italian readers to know where the Lazio region is without having to resort to an atlas. The major outbreak in northern Italy is well-known, so giving a rough location (e.g. surrounding Rome) would be helpful.

Some of the sentences are quite long, which makes reading hard. Running the manuscript through an online readability test should highlight the problem and point out specific sentences to split.

References are important to maintain continuity of knowledge. Online resources are particularly prone to "link rot" and knowing the date and the link (url) in accessing the reference are useful. Conversely, it is helpful to describe the reference in the References and not simply put in a link.

Author Response

  • In lines 67-68, the Lazio region has been better explained in the text, giving importance to Institute location in Rome.
  • The manuscript underwent to on-line readability and several too long sentences have been shortened to make the text more fluid in reading. Further English revision has been achieved to correct minor style or grammatical errors.
  • As it can be checked in references, all links were specifically cited with URL and date of consulting.

Reviewer 3 Report

Nice summary of your institution experience. 

Maybe include comment ans citation of COVID-guidelines/recommendations like ESMO-EURACAN and compare them with the performance in your institution. 

It´s not clear for me if radiotherapy is applied at your institution or patients are refered to other institutions. Maybe expand the explanations regarding changes due to COVID to other disciplines: the chart for surgery indications could include chemo or radiotherapy indications during pandemia, or changes like use of GCSF or change to oral scheldules for exemple. 

Author Response

Thank you very much for the suggestion to consider ESMO/EURACAN recommendation. In reality, at the time of the first draft of the paper, the purpose of the study was to focus only on organization flowchart to manage sarcoma patients but ESMO GL were not published yet. In this version, we have considered our surgical indication and treatment prioritization at light of ESMO GL (see the paragraph “Sarcoma unit approach” where also medical and radiation oncology has been integrated) and we found a satisfactory adhesion and behavior correspondence between our Sarcoma Unit and ESMO recommendations, as reported in discussion.